# An intercomparison of four gridded precipitation products over Europe using an extension of the three-cornered-hat method

Llorenç Lledó[1], Thomas Haiden[2], and Matthieu Chevallier[2]

[1]ECMWF, Bonn, Germany
[2]ECMWF, Reading, UK

**Correspondence:** Llorenç Lledó (llorenc.lledo@ecmwf.int)

**Abstract.** Precipitation is arguably one of the most relevant surface variables impacting human lives on the planet, but global-coverage, high-resolution and good-quality observations are not readily available. In particular, gridded observational datasets are much needed for model development and forecast quality assessment. Here we compare the quality of four types of gridded precipitation products over Europe, namely: a rain-gauge interpolation; a satellite-derived product; a radar composite; and a reanalysis. Each product has its own strengths and weaknesses, and since each precipitation estimate uses different measuring techniques, we can employ a triangulation method to estimate the error variance of each product with respect to the unknown true values. Results show that: a) the satellite product has limited quality over Europe and may be problematic to use in quantitative forecast evaluation and diagnostics; b) the radar composite has spurious features that need to be considered when used in verification; c) all products struggle in topographically complex areas; d) the rain-gauge interpolation is not free of errors, despite rain gauges being often treated as ground truth in the literature; and e) the reanalysis dataset produces in some cases the best available estimates, in particular over the European near-coastal waters.

## 1 Introduction

Surface precipitation affects many aspects of human life on Earth: rain, droughts and floods can threaten human lives and infrastructures, and impact agriculture and energy production, biodiversity, forest fires, air quality, availability of drinking water, health, businesses, and everyday lives. Unsurprisingly, precipitation forecasts are an essential part of weather prediction, and the importance of measuring precipitation globally has been widely recognized (Hou et al., 2014). In particular, gridded precipitation observations are needed to assess socio-economic impacts of precipitation hazards, better understand physical processes, calibrate numerical models, or verify weather forecasts.

While nowadays many observational precipitation products are available (e.g. see Beck et al. (2019) for a list of 26 alternatives), their spatial coverage, horizontal resolution, temporal sampling, and overall quality vary widely. Each product has its own strengths and weaknesses: for instance, some products have very good quality but are only available over land or for certain countries; other datasets have global coverage at the cost of lower resolution; some datasets include rain estimates over the oceans, where in-situ observations are not available; and some products are designed to be available close to real-time, while other products put more emphasis on quality control and homogeneity.

In view of this variety of products, it is crucial to assess both their uncertainty and the suitability for specific purposes. This study was guided by the need to verify kilometre-scale global forecasts produced under the framework of the Destination Earth programme (Hoffmann et al., 2023). In the case of forecast verification, the quality of the observational datasets affects the score results and can potentially alter the conclusions reached (Bowler, 2008; Candille and Talagrand, 2008; Duc and Saito, 2018; Bessac and Naveau, 2021; Ramon et al., 2023). In practical terms, high-resolution observations with a wide coverage over

land and oceans should be preferred for the verification of global kilometre-scale forecasts. Precipitation is highly variable both in space and time, and its highly skewed distribution renders time or space aggregates non-representative of the most extreme conditions occurring within a grid box.

    In this work we compare the quality of four independently derived precipitation products over Europe (see section 3.1). We employ a triangulation technique to estimate the error variance of each product (described in section 3.3). This method

relies on the assumption that the precipitation estimates are made using independent observation and retrieval techniques. Here, we have extended the method to allow for a certain degree of inter-dependence in the datasets. Therefore, we review the existing measurement techniques in section 2 to understand their strengths and shortcomings. Results and conclusions follow in sections 4 and 5.

## 2   Four alternative ways of estimating precipitation

Several measurement techniques are available to estimate surface precipitation (Prigent, 2010; Sun et al., 2018). Rain gauges are the most direct way of observation, and if well maintained and installed they produce good-quality records. However, gauges have problems measuring when there are high winds, or when the precipitation is not in liquid phase, and they produce estimates that are only valid at the local scale. Moreover, their installation and maintenance are costly and therefore there is a limited number of such stations, mostly concentrated around populated areas. Gridded products can be produced from gauge

networks, and their quality will be dictated by the number of stations within each grid cell.

    Remote sensing provides an alternative to in-situ measurements of surface precipitation. Weather radars can measure precipitation in the surroundings by sending horizontal beams of electromagnetic waves and measuring the energy reflected by water suspended in the air. Several types of radars with different frequency bands are operating in many countries. Radars produce high-resolution information down to kilometre-scale, but they have known issues: beam attenuation (quality degrades when

moving away from the radar location); beam blocking by topography; the indirect nature of the measurement which introduces errors when transforming measured reflectivity to precipitation rates (Z-R relationship); spurious echoes and clutter due to wind turbines, bird migrations, planes, etcetera; or the inability to measure close to ground level, therefore confusing virga for precipitation or missing low-level precipitation. Weather radars are also a costly infrastructure, and they are only available for populated areas and in countries that can afford it. Gridded products can be made by combining information from multiple

radars.

    Satellite-mounted remote sensing devices allow measuring precipitation from space in regions where traditional observations or weather radars are not available, e.g. over the oceans or in remote areas. Many types of satellites and instruments can be

**Table 1.** Main characteristics of the observational datasets employed.

|  | E-OBS | IMERG | OPERA | ERA5 |
|---|---|---|---|---|
| Product type | Rain gauges | Satellite | Radar | Reanalysis |
| Spatial resolution | 0.1 degrees | 0.1 degrees | ~2 km | 0.25 degrees |
| Coverage | Europe, only land | Global* | Europe, land and offshore | Global |
| Time sampling | Daily | 30 min. | 15 min. | hourly |
| Period | 1950–present | 2010–Sep 2021 | March 2013–present | 1940–present |

\* Fewer instruments available above 60ºN.

employed. Infrared imagers on geostationary satellites can estimate convective precipitation through brightness temperatures and cloud top height information. Low-Earth-orbiting satellites can also estimate precipitation by measuring passive microwave radiation or by employing active microwave instruments (e.g. on-board weather radars). The main sources of uncertainty for satellite-derived precipitation observations are: the indirect nature of the observations; the sparse or intermittent coverage of polar-orbiting satellites; the inhomogeneity of the techniques employed over land and sea, e.g. by passive microwave imagers; difficulties to detect warm-cloud (low-level) precipitation; and the algorithms needed to combine the available information from multiple instruments and satellites.

Finally, reanalyses combine numerical weather prediction models and data-assimilation techniques to estimate a gap-free and physically consistent state of the atmosphere. Current global reanalyses have coarser resolutions than other state-of-the-art precipitation observation systems, but they offer an additional perspective of surface precipitation. Precipitation estimates from reanalyses are not derived directly from precipitation measurements, but by observations of other atmospheric quantities such as temperature, humidity, or atmospheric circulation, which in turn shape precipitation in the model. Precipitation estimates from reanalyses inherit the weaknesses of the models employed (e.g. they rely on parametrizations), and have difficulties representing small-scale convective precipitation, but can do a good job reproducing larger-scale stratiform precipitation.

## 3 Data and methods

### 3.1 Datasets

Four gridded datasets covering Europe have been employed. Table 1 summarises the main characteristics of each of them. While many other gridded precipitation datasets are available, for this study we have selected one product for each type of measurement technique to guarantee independence of products as much as possible.

#### 3.1.1 E-OBS

E-OBS (Cornes et al., 2018) is a gridded product produced by KNMI and derived from in-situ rain-gauge measurements over Europe. As such, it only provides values over land. The station density within each grid cell is variable both in space and time,

impacting the quality of the precipitation estimates. We have employed version 26.0e of this dataset. As an in-situ product, it is sometimes used as a reference for calibrating or verifying other types of measurements, although that is only valid in areas of high station density. Moreover, this product is not free of errors. For instance, the daily accumulation periods of some gauges do not correspond to accumulations at midnight. All our analyses are agnostic and do not assume that one product is an error-free reference used to benchmark others.

### 3.1.2 IMERG

The Integrated Multi-satellitE Retrievals for Global Precipitation Mission (IMERG), (Huffman et al., 2023, 2020) is a satellite-derived product combining information from passive and active sensors aboard both geostationary and polar-orbiting satellites. Its main strength is that it provides global coverage (including oceans) with a 30-minute time sampling on a 0.1 degree grid. The IMERG algorithm is complex and merges infrared temperatures, passive microwave, and active radar soundings. We employ the Final Run IMERG version 07B product, which has been quality controlled and scaled so that monthly totals match those of rain gauges, where available.

### 3.1.3 OPERA

The Operational Program on the Exchange of Weather Radar Information (OPERA), (Huuskonen et al., 2014; Saltikoff et al., 2019) produces a weather radar composite combining information from around 200 radars from 25 European countries. This radar network is very inhomogeneous in terms of radar technologies employed in different countries (bands, scanning strategy, etc. . . ), but the Opera data center (ODC, also known as Odyssey) produces the composites from individual information of each radar, which minimizes the differences due to post-processing. The hourly total precipitation composite is available on a grid of 2 km resolution every 15 minutes. As a very basic measure of quality control, we have set any values above 300 mm/h as invalid.

### 3.1.4 ERA5

ERA5 (Hersbach et al., 2023, 2020) is a state-of-the-art global reanalysis produced by ECMWF. By employing a frozen numerical weather prediction model and data assimilation scheme it recreates the state of the atmosphere during the past decades. A reanalysis constrains the solution to be physically consistent and close to existing observations available. Therefore, it produces accurate estimates of precipitation even in regions without direct observations. Indeed, ERA5 does not assimilate precipitation observations other than the NCEP stage IV precipitation over the US. This means that precipitation estimates from ERA5 in Europe derive mainly from other atmospheric quantities constrained by observations (e.g. humidity, geopotential height, winds. . . ) and their relationship to precipitation in the model. Total precipitation in ERA5 includes both parametrized and large-scale precipitation in forms of rain and snow, and takes into account some of the precipitation evaporation.

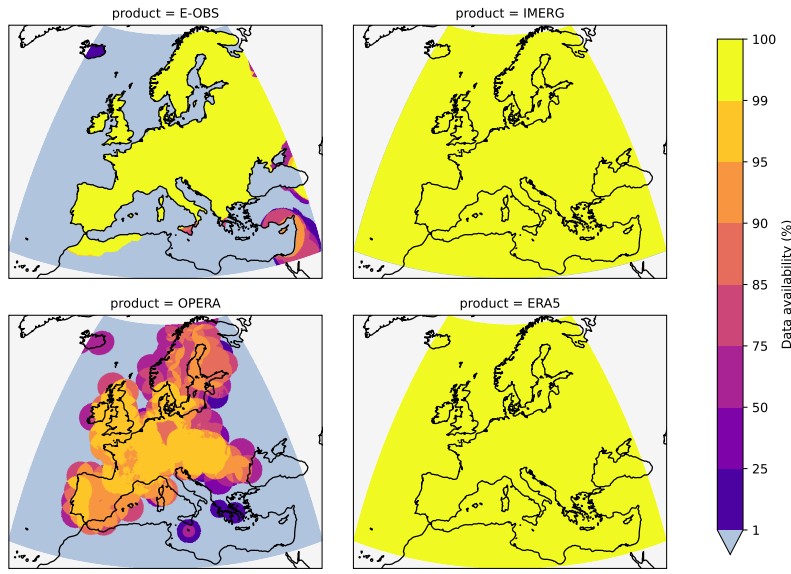

**Figure 1.** Data availability for each product, indicated as a percentage of the whole period.

## 3.2 Data collocation

In order to perform the comparative assessment, all the data has been collocated in a common space and time grid. First the data has been accumulated to daily total precipitation (00:00–24:00UTC). Then each dataset has been interpolated to a 0.1ºx0.1º latitude-longitude grid. In particular, the OPERA radar data has been interpolated from a finer grid using a conservative inter-polation (in particular the "average" method of the gdalwarp utility), and the ERA5 reanalysis has been interpolated from a coarser grid with a nearest-neighbour method. The E-OBS and IMERG products were already on a 0.1º grid.

The period employed covers October 2016 to September 2021, amounting to 5 years or 1826 days. The data has been cropped to a European domain covering 20ºW to 40ºE and 30ºN to 70ºN. However, some of the assessed products do not cover the whole domain, and a full comparison is only possible for those grid points where all four products are available. Figure 1 shows a map of data availability (in %) for each product. E-OBS is only available over land. The radar composite covers only a limited number of countries with an additional offshore buffer. ERA5 and IMERG cover all the region and for all times,

although IMERG has a reduced satellite coverage above of 60ºN.

The assessment is performed separately at each grid point. To ensure that the statistics (means, variances, covariances) computed for each product are comparable, only the days that are available for all four products are considered. Therefore, the exact periods at each location can slightly differ. ERA5, IMERG and E-OBS are available almost all the time, so the dates used at each grid point are dictated by the availability of OPERA data. Additionally, we have discarded any grid point with

less than 75% of data availability, ensuring sample sizes of at least 1370 days. This is within the range of 500 to 5000 samples recommended by Sjoberg et al. (2021) for the 3CH method.

## 3.3 Three-cornered-hat error estimation

The three-cornered-hat (3CH) method was originally developed to estimate random errors of atomic clocks (see Sjoberg et al. (2021) for a comprehensive review of the method). The 3CH and the related triple collocation method have been used in the areas of weather and climate to estimate observational errors of wave height (Janssen et al., 2007), sea surface temperature (O'Carroll et al., 2008), precipitation (Roebeling et al., 2012; Massari et al., 2017; Alemohammad et al., 2015), surface wind (Abdalla and De Chiara, 2017), soil moisture (Gruber et al., 2016) or radio occultation measurements (Anthes and Rieckh, 2018).

The goal is to quantify the error incurred by an observation system $A$ when trying to estimate some (unknown) true values $\tau$. We define the (additive) error of $A$ as: $\mathcal{E}_A = A - \tau$. The method relies on decomposing the variance of the difference of two independent observation systems $A$ and $B$ as:

$$\text{Var}(A - B) = \text{Var}(\mathcal{E}_A - \mathcal{E}_B) = \text{Var}(\mathcal{E}_A) + \text{Var}(\mathcal{E}_B) - 2\text{Cov}(\mathcal{E}_A, \mathcal{E}_B) \tag{1}$$

which can then be approximated by the sum of the error variance of each product:

$$\text{Var}(A - B) \approx \text{Var}(\mathcal{E}_A) + \text{Var}(\mathcal{E}_B) \tag{2}$$

The approximation is only valid if the observational errors of product $A$ and $B$ are uncorrelated, i.e. the observations are derived independently. For example, the second panel in figure 2 shows the variance of differences between the gauge product and the radar composite. If we assume that radar and gauges measure the same unknown quantity $\tau$ employing different techniques which should produce uncorrelated errors, we can interpret figure 2 as the sum of the individual error variances of both products. Areas with large values in figure 2 are caused by errors in either product $A$ or $B$ or both (notice that variances cannot be negative by definition, so there is no room for error compensation).

At this point we cannot say what proportion of the difference variance is attributable to each product error. To that end, if we have a third independent observation system $C$, we can repeat the above steps with differences between $A$ and $C$, and $B$ and $C$ to obtain a linear system of three equations and six unknowns (three variances and three covariances) that can be solved to obtain a quantitative result:

$$\begin{cases} \text{Var}(A - B) = \text{Var}(\mathcal{E}_A) + \text{Var}(\mathcal{E}_B) - 2\text{Cov}(\mathcal{E}_A, \mathcal{E}_B) \\ \text{Var}(A - C) = \text{Var}(\mathcal{E}_A) + \text{Var}(\mathcal{E}_C) - 2\text{Cov}(\mathcal{E}_A, \mathcal{E}_C) \\ \text{Var}(B - C) = \text{Var}(\mathcal{E}_B) + \text{Var}(\mathcal{E}_C) - 2\text{Cov}(\mathcal{E}_B, \mathcal{E}_C) \end{cases} \tag{3}$$

Assuming that the three error covariance terms are small and can be neglected, the system of equations can be solved for the error variances from the three collocated datasets:

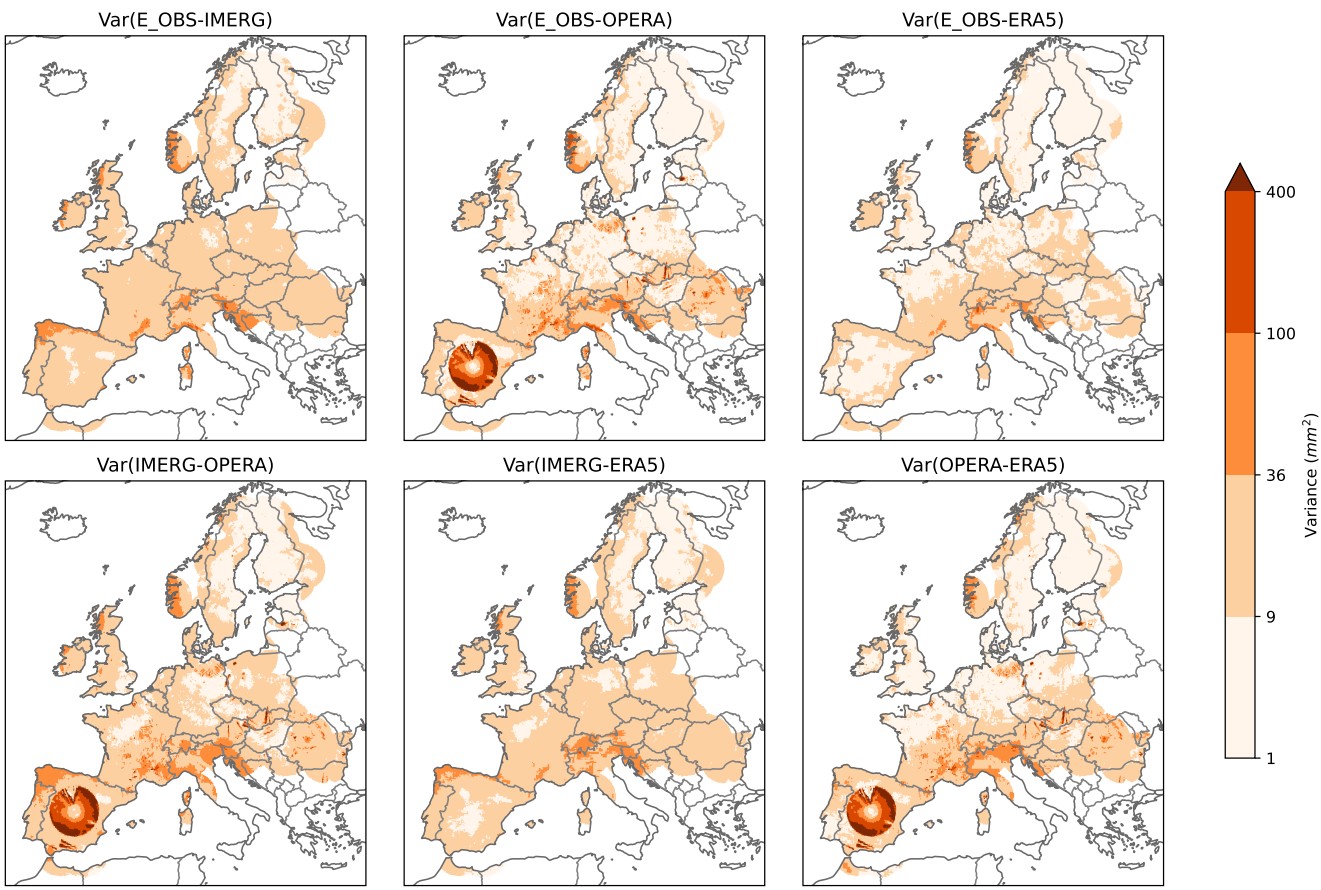

**Figure 2.** Variance of the differences between pairs of products. A circular pattern around Madrid can be clearly seen for all panels involving the radar composite, uncovering a clear problem in that dataset even before performing the 3CH analysis.

$$
\begin{cases}
\mathrm{Var}(A - B) = \mathrm{Var}(\mathcal{E}_A) + \mathrm{Var}(\mathcal{E}_B) \\
\mathrm{Var}(A - C) = \mathrm{Var}(\mathcal{E}_A) + \mathrm{Var}(\mathcal{E}_C) \\
\mathrm{Var}(B - C) = \mathrm{Var}(\mathcal{E}_B) + \mathrm{Var}(\mathcal{E}_C)
\end{cases}
\tag{4}
$$

### 3.4 Four-cornered-hat error estimation

In our case we have identified four observational systems, therefore we can constrain the computation a bit more. In the general case of $N$ datasets there are $\binom{N}{2}$ unique pairs of products for which we can write an equation like eq.1. The resulting system of linear equations is under-determined, with $\binom{N}{2}$ equations, $N$ unknown variances and $\binom{N}{2}$ unknown covariances. For $N = 4$ we have six equations and ten unknowns:

$$\begin{cases}
\text{Var}(A - B) = \text{Var}(\mathcal{E}_A) + \text{Var}(\mathcal{E}_B) - 2\text{Cov}(\mathcal{E}_A, \mathcal{E}_B) \\
\text{Var}(A - C) = \text{Var}(\mathcal{E}_A) + \text{Var}(\mathcal{E}_C) - 2\text{Cov}(\mathcal{E}_A, \mathcal{E}_C) \\
\text{Var}(A - D) = \text{Var}(\mathcal{E}_A) + \text{Var}(\mathcal{E}_D) - 2\text{Cov}(\mathcal{E}_A, \mathcal{E}_D) \\
\text{Var}(B - C) = \text{Var}(\mathcal{E}_B) + \text{Var}(\mathcal{E}_C) - 2\text{Cov}(\mathcal{E}_B, \mathcal{E}_C) \\
\text{Var}(B - D) = \text{Var}(\mathcal{E}_B) + \text{Var}(\mathcal{E}_D) - 2\text{Cov}(\mathcal{E}_B, \mathcal{E}_D) \\
\text{Var}(C - D) = \text{Var}(\mathcal{E}_C) + \text{Var}(\mathcal{E}_D) - 2\text{Cov}(\mathcal{E}_C, \mathcal{E}_D)
\end{cases} \tag{5}$$

In order to find a unique solution we need $N$ additional constraints. For the 3CH ($N = 3$) all three covariance terms were set to zero. For $N = 4$ we can set any four of the six covariance terms to zero, and then the two remaining covariance terms will be explicitly computed by the method.

Assessing which two covariance terms should be computed is a science-informed but subjective matter. The independence of two observational products reduces to identifying confounding variables that could have an impact on the measurements on both measuring techniques. Notice that independently derived products do not directly guarantee uncorrelated errors. For example, if two measuring approaches are sensitive to rain drop size, or two products have difficulties to identify low-level precipitation, that will result in correlated errors in the data. Small sample sizes can also result in non-zero covariances due to sampling randomness. In our setting, the following error orthogonality assumptions have been made:

– Satellite and gauge errors are considered independent since measurement techniques are very different in nature. Although the final IMERG product is bias adjusted with monthly total precipitation of some gauge stations, the variability of daily total values is only weakly affected by that adjustment. Therefore, we set $\text{Cov}(\mathcal{E}_{IMERG}, \mathcal{E}_{EOBS}) = 0$

– Radar and gauge errors are considered independent since measurement techniques are very different in nature. OPERA does not use rain gauges to calibrate the composites. Therefore, we set $\text{Cov}(\mathcal{E}_{OPERA}, \mathcal{E}_{EOBS}) = 0$

– ERA5 and gauge errors are considered independent since ERA5 is not assimilating any rain gauge information beyond the US. Therefore, we set $\text{Cov}(\mathcal{E}_{ERA5}, \mathcal{E}_{EOBS}) = 0$

– ERA5 and radar errors are independent, since ERA5 is not assimilating any radar data over Europe. Therefore, we set $\text{Cov}(\mathcal{E}_{ERA5}, \mathcal{E}_{OPERA}) = 0$

– ERA5 assimilates radiances from many satellites, including microwave imagers, microwave sounders and infrared imagers (see figure 5 in Hersbach et al. (2020)). Some of those instruments (e.g. the GPM microwave imagers) are also used by the IMERG algorithm (Huffman et al., 2020). Therefore, we have decided to explicitly compute $\text{Cov}(\mathcal{E}_{IMERG}, \mathcal{E}_{ERA5})$.

– Some GPM satellites used in IMERG incorporate a dual-band weather radar on-board. Although the overall characteristics of this on-board radars is very different from conventional ground-based radars, some aspects such as the equations

employed to convert radar reflectivity to precipitation rates are common to both estimate techniques. Therefore, we have decided to explicitly compute $\mathrm{Cov}(\mathcal{E}_{IMERG},\mathcal{E}_{OPERA})$.

Some extensions of the 3CH that take advantage of more than three collocated datasets have already been proposed in the literature. On one hand Sjoberg et al. (2021) already showed how $N$ datasets can be used to obtain $\binom{N-1}{2}$ different estimates of $\mathrm{Var}(\mathcal{E})$ for each product by selecting distinct triplets of datasets and employing the 3CH on each subset. Alternatively, Pan et al. (2015) present the equations of 3CH (albeit referring to triple collocation) assuming uncorrelated errors for all pairs of products. This turns the problem into an over-determined system which is then solved by ordinary least squares. Finally,
Zwieback et al. (2012) or Vogelzang and Stoffelen (2021) present a quadruple-collocation solution which is conceptually identical to ours but based on the triple-collocation equations, which rely on slightly different assumptions. The strength of the approach we present is that the error orthogonality assumptions are underpinned by domain knowledge, resulting in a better approximation of the $\mathrm{Var}(\mathcal{E})$ values (Zwieback et al., 2012).

## 4   Results

As a first step to intercompare the four datasets, the difference in the daily total mean precipitation is presented in figure 3 for each pair of products. The largest differences can be seen over the ocean, where OPERA is drier than IMERG and ERA5 by 2 to 5 mm, and IMERG is wetter than ERA5 by 1 to 2 mm. Over land, ERA5 and IMERG have a better agreement, with important differences concentrated over high mountain ranges only. Both products are wetter than E-OBS and OPERA, although OPERA is the driest product in most locations. While these difference plots allow spotting spatial patterns in the data
(e.g. beam attenuation in all OPERA panels, or systematic biases over Poland for E-OBS) and getting an idea of the uncertainty range, the true values are unknown, and hence we cannot judge which of the products yields a closer estimate of the true mean precipitation.

    Under the assumptions made in the previous section, the 4CH method allows estimating the error variances of each product and therefore ranking the products from highest to lowest. Figure 4 shows the 4CH error variances of each product (only
available over the grid points covered by all four products). The IMERG product shows the largest errors overall, which tend to be highest in coastal areas of the Iberian Peninsula and the British Isles, and around the Alps. The OPERA composite also has very high error variances in some regions, clearly related to issues in particular radars, e.g. the radars in Madrid or Copenhagen, but in other areas such as Germany the quality is very good. Clutter can also be seen in some areas of eastern Europe. It is reassuring to see that the method can recreate the spatial error patterns produced by the radars without any knowledge of the
radar locations. The E-OBS product emerges as a good product in almost all of the continent, although it has some quality issues in areas such as the Balkans. The ERA5 dataset, despite being a lower-resolution product, shows errors close to those of E-OBS, and even lower in some regions such as Brittany or the British Isles (see figure A1 for a plot of the ranking of each product at each grid point). ERA5 has less quality in mountainous areas such as the Alps. Some localized spots in figure 4 have negative variance values. Those can be mainly seen for E-OBS over the Iberian Peninsula and in the north-east of Italy,
with a mean value close to -1.6 mm$^2$ over those locations. Those values are unphysical, since error variances must be positive

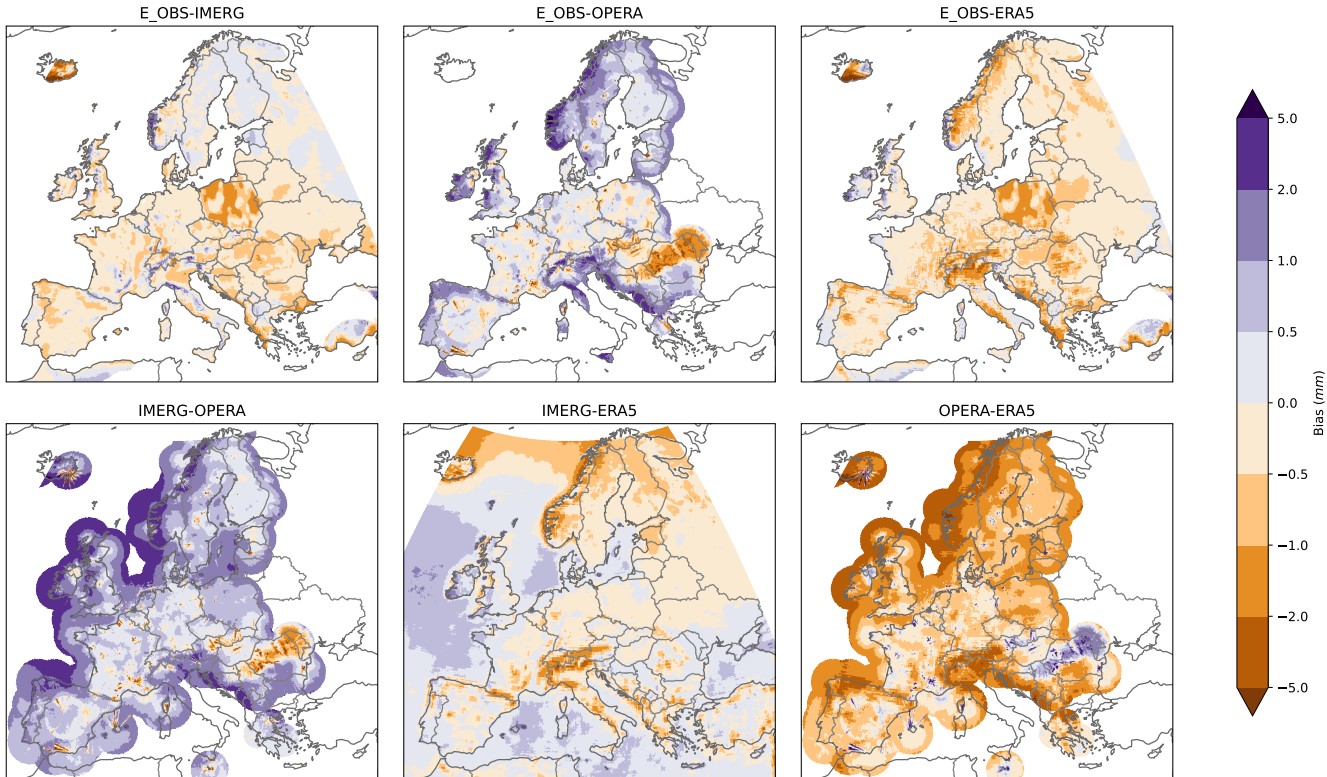

**Figure 3.** Mean of the differences between pairs of products.

by definition, and result from the assumptions made in section 3 not being entirely correct in those grid points (see section 4e in Sjoberg et al. (2021) and section 2.5 in Pan et al. (2015) for detailed discussions of when that occurs). Indeed, the error covariance between IMERG and ERA5 has strong positive values in those regions (see figure 5), which are then subtracted during the computation of the variance estimate of E-OBS (see equation 3a in Sjoberg et al. (2021)). While some authors have proposed ways forward to constrain the 3CH solutions to remain positive-definite (e.g. see Premoli and Tavella (1993)), we have not explored this avenue, because the issue is minor and restricted to few locations in our case. The corresponding error correlations (see figure A2) are mostly positive and not particularly small, especially for the two remote sensing products.

Since convective and stratiform precipitation have very different characteristics in terms of spatial patterns, intensity, duration, or height in the atmosphere, it is reasonable to ask if the four products studied here represent well these two distinct types of precipitation. To answer that, we have stratified the results into two seasons, an extended winter (NDJFM, predominantly stratiform precipitation) and extended summer (MJJAS, mostly convective precipitation). The results in figure 6 show that E-OBS, OPERA and ERA5 have better accuracy in winter than in summer, whereas IMERG has better performance in summer than in winter. This confirms that IMERG is better at sensing convective precipitation events than stratiform precipitation, whereas the contrary can be said for the other products. Still, IMERG is not better than the other three products in

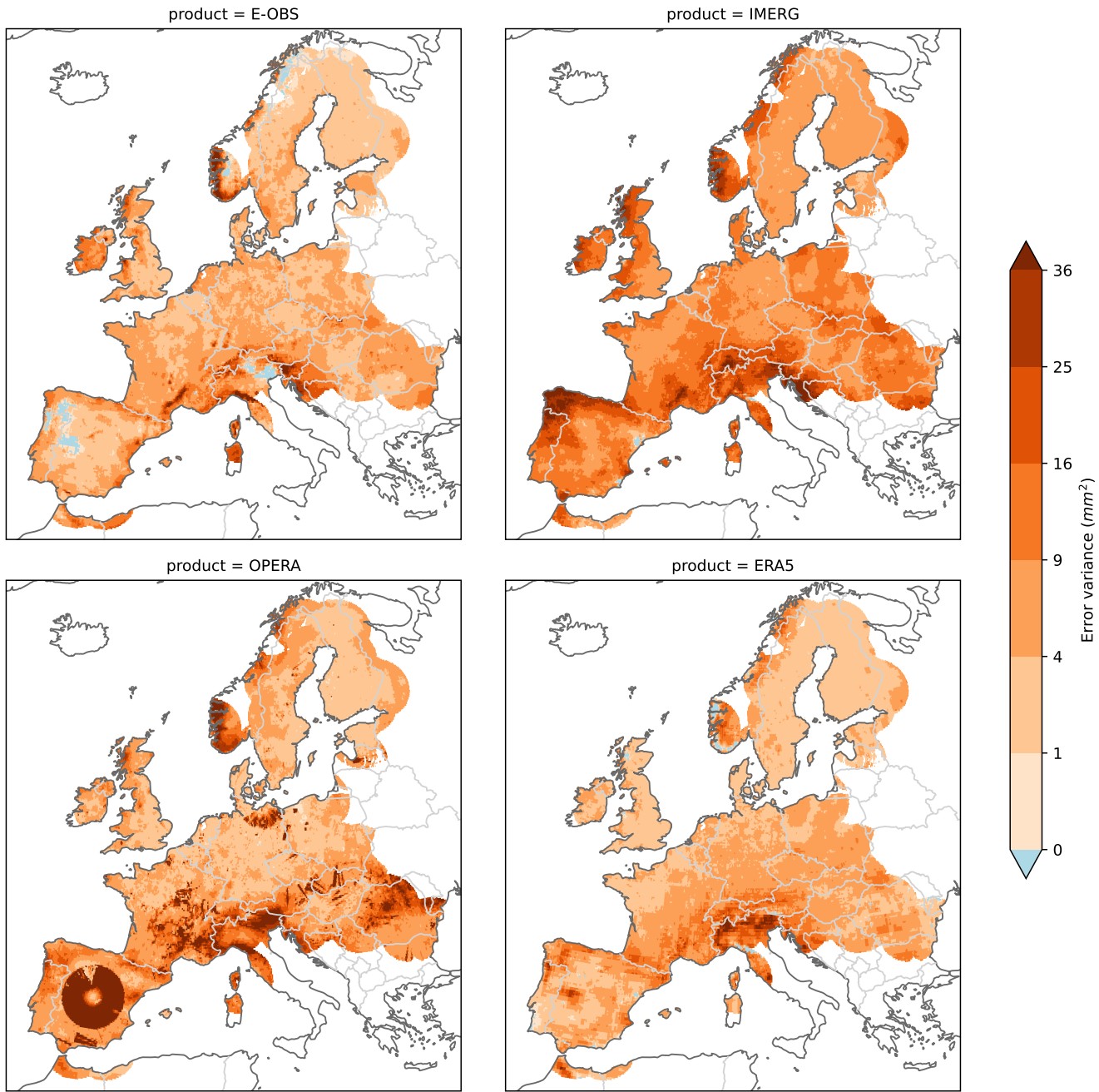

**Figure 4.** 4CH-derived error variances for the four products.

summer. An interesting aspect to notice in figure 6 is that the quality issues we saw in figure 4 for OPERA almost disappeared in the winter/summer stratifications. A plausible explanation is that some outliers that are only present in some days of April

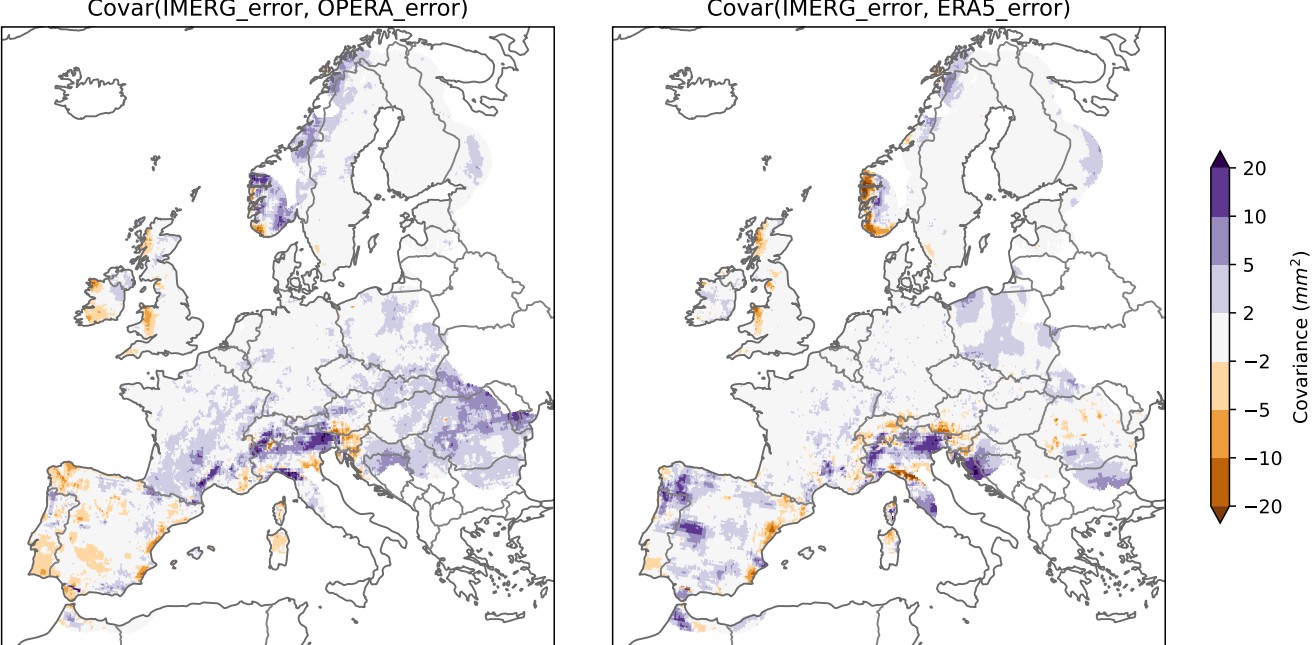

**Figure 5.** 4CH-derived error covariances for the two explicitly computed pairs of products.

or October (therefore excluded in both composites) had a strong influence on the overall result (see section 4 of Sjoberg et al. (2021)).

## 5 Conclusions

The choice of a verification dataset for global and high-resolution precipitation forecasts is challenging. From a technical perspective IMERG is a good choice because it covers the oceans, and it produces a relatively high resolution and mostly gap-free dataset. However, we have seen that its quality over Europe is not particularly good compared to existing alternatives. Moreover, its quality over the ocean is also not good (see figure A3). Both E-OBS and the OPERA radar composites provide a better alternative over European land. The OPERA data is a higher resolution product, and although it might be biased with

respect to in-situ measurements, with a bit of additional quality control it might be the best way to go for verifying high-resolution simulations over Europe. Both products have the limitation of not providing information over the oceans, which might be a problem to obtain robust results (given that oceans cover two thirds of our planet) or for applying spatial verification techniques that rely on neighbourhood analyses. The only remaining alternative might then be using modelled products such as ERA5. Our results show that the quality of ERA5 is not far from that of E-OBS in Europe, especially in winter (in agreement

with Lavers et al. (2022)). But since ERA5 is produced with a specific version of the IFS model, comparing IFS forecasts to ERA5 does not provide a very stringent evaluation, as forecast and analysis share very similar model limitations and systematic

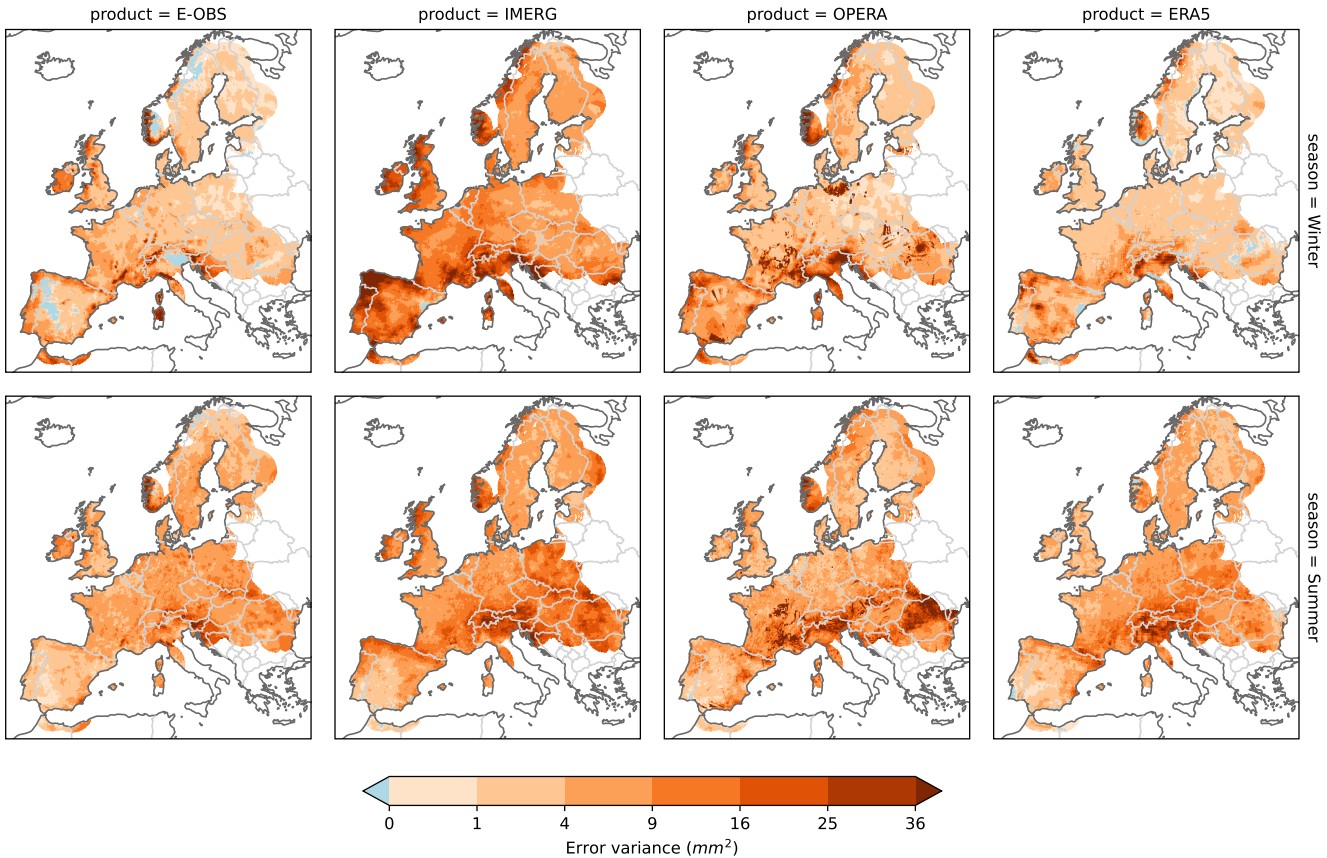

**Figure 6.** 4CH error variances in winter (top row) and summer (bottom row) for each observational product.

biases. Higher-resolution modelled datasets such as regional reanalyses could provide better results than ERA5, but we have not explored this option here. In short, the promise of a global-coverage, high-resolution and good-quality precipitation dataset measured from space is not yet fully realized, and therefore verifying global precipitation forecasts against reanalysis datasets is still a justified alternative. All the conclusions above are based on estimates of error variance obtained with an extension of the three-cornered-hat method, which in turn relies on a set of error covariance assumptions. Those assumptions were carefully selected based on physical considerations of the different observational systems but have not been independently validated. The error variance estimates can differ from the true error variances whenever the error orthogonality assumptions are not fulfilled.

Extending this study to other regions would be interesting. Specifically, the IMERG dataset is thought to be of better quality in the tropics due to the predominance of convective precipitation, and it has already been used there to assess the quality of convective-permitting simulations (Becker et al., 2021). Unfortunately, an analysis such as the one we presented is only possible in areas with a good radar and gauge network. The US would be an ideal place to apply the 4CH method, but unfortunately there the ERA5 assimilates the stage IV (radar and gauges) product, making our assumptions invalid.

This study has only dealt with error variances, and therefore does not directly address quality aspects of the precipitation products that might be relevant for extreme events, such as performance under heavy rain (see e.g. Sun et al. (2018)). Additional analyses showing other quality aspects of the IMERG dataset in Europe can be found in Navarro et al. (2019) or O et al. (2017), whereas the quality of IMERG over the US has also been studied in Beck et al. (2019). These studies take one reference dataset as the true values and evaluate other products. However, our agnostic analysis has confirmed that gauge-based products are not free of errors either.

*Code and data availability.* All the analyses were done with Python and Xarray (Hoyer and Hamman, 2017). A Jupyter Notebook containing all the analyses can be found at https://github.com/lluritu/4CH_precip_comparison. E-OBS version 26.0e can be obtained from the Coperni-cus Climate Change Service at https://surfobs.climate.copernicus.eu/dataaccess/access_eobs.php. The GPM IMERG v07B can be accessed from NASA GES DISC at https://doi.org/10.5067/GPM/IMERG/3B-HH/07. OPERA data is available from EUMETNET (www.eumetnet.eu/opera). ERA5 was downloaded from the Copernicus Climate Change Service (C3S) (2023) at https://doi.org/10.24381/cds.adbb2d47.

**Appendix A**

This appendix contains three additional figures. Figure A1 shows a ranking of the four products according to the 4CH analysis, from lowest to highest error variance at each grid point, which helps realizing which datasets are more accurate overall. Figure A2 displays the error correlations derived from the error covariances and variances in figures 4 and 5. It is worth mentioning that there is no formal guarantee that this derived correlations remain in the [-1,1] range, and even they cannot be computed when variances are zero or negative. Finally figure A3 shows an additional 3CH analysis including only IMERG, OPERA and ERA5. In this case the errors are assumed uncorrelated. We can see that IMERG and OPERA datasets struggle to represent precipitation over the oceans, whereas ERA5 has negative values there. This indicates that IMERG and OPERA errors are indeed correlated over the ocean. The results over land are comparable to those in figure 4, although the errors for IMERG seem to be a bit lower with this simpler hypothesis.

*Author contributions.* Llorenç Lledó: conceptualization, methodology, data curation, software, visualization, writing - original draft prepa-ration. Thomas Haiden: conceptualization, methodology, writing - review and editing. Matthieu Chevallier: writing - review and editing.

*Competing interests.* The authors declare that they have no conflict of interest.

*Acknowledgements.* The work presented in this paper has been produced in the context of the European Union's Destination Earth Initiative and relates to tasks entrusted by the European Union to the European Centre for Medium-Range Weather Forecasts implementing part of

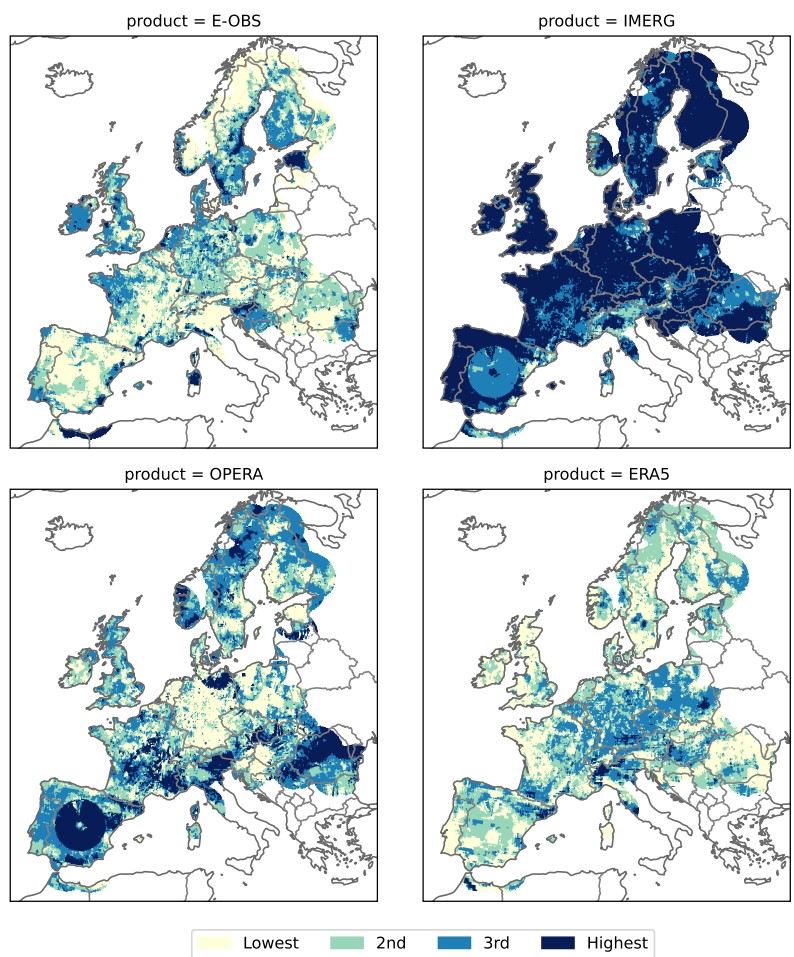

**Figure A1.** Ranking of the four products according to 4CH error variance.

this Initiative with funding by the European Union. Views and opinions expressed are those of the author(s) only and do not necessarily reflect those of the European Union or the European Commission. Neither the European Union nor the European Commission can be held responsible for them. The authors want to thank Philippe Lopez and Tobias Becker for assistance with the pre-processing of OPERA and IMERG datasets respectively, and Estíbaliz Gascón for assessing the quality of IMERG for specific case studies. We acknowledge the E-OBS dataset from the EU-FP6 project UERRA (https://www.uerra.eu) and the Copernicus Climate Change Service, and the data providers in the ECA&D project (https://www.ecad.eu). The GPM IMERG data were provided by the NASA Goddard Space Flight Center's Precipitation Measurement Missions Science Team and Precipitation Processing System, which develop and compute the GPM IMERG as a contribution to GPM, including data archiving at the NASA GES DISC. ERA5 was downloaded from the Copernicus Climate Change Service (C3S) (2023). The results contain modified Copernicus Climate Change Service information. Neither the European Commission nor ECMWF is responsible for any use that may be made of the Copernicus information or data it contains.

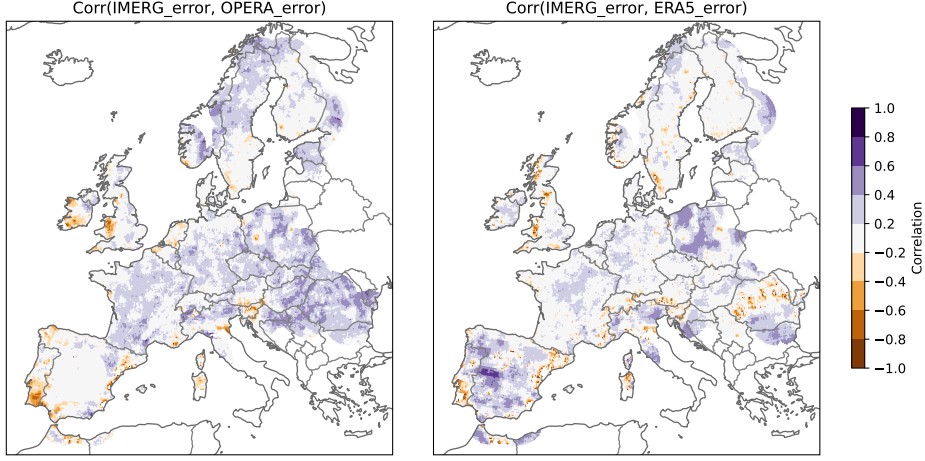

**Figure A2.** 4CH-derived error correlations for the two explicitly computed pairs of products.

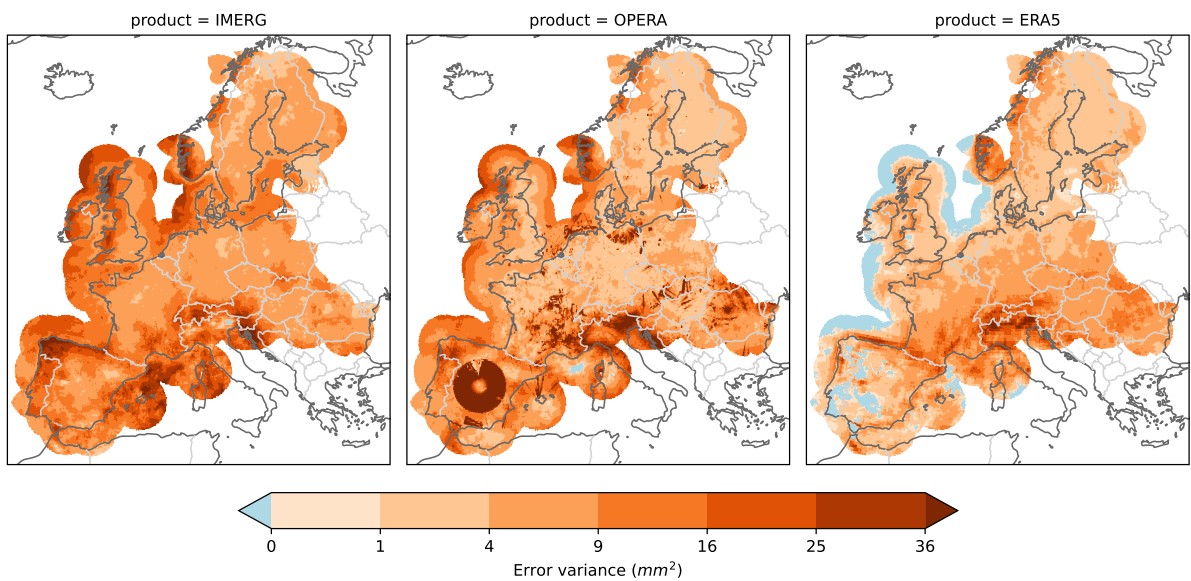

**Figure A3.** Error variance of the three products that cover the near-coastal European waters, under the assumption of uncorrelated errors.

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
