# Peer review of "An intercomparison of four gridded precipitation products over Europe using an extension of the three-cornered-hat method"

_EGUsphere, 2024_

## Referee Comment (RC1)

3 April 2024

Review of *An intercomparison of four gridded precipitation products over Europe using the three-cornered-hat method* by L. Lledó, T. Haiden, and M. Chevallier for EGUsphere.

Richard Anthes
anthes@ucar.edu

This is an excellent paper that applies the three-cornered hat (3CH) and four-cornered hat (4CH) methods to estimate the uncertainties (random error variances) of four precipitation datasets. It is acceptable for publication after the authors consider some relatively minor changes that would improve the clarity of the paper.

A novel aspect of the paper is to use the 4CH method to compute two of the error covariance terms that must be neglected in 3CH error variance estimates. Sjoberg et al. (2021) show that N datasets lead to (N - 1)(N - 2)/2 unique error variance estimates for each dataset when using the 3CH method. Furthermore, under the assumption of zero error covariances among all the datasets, a single error estimate for each dataset can be computed using all N datasets simultaneously (N-cornered hat method, a generalization of the 3CH method), which is identical to the mean of all the individual estimates. This paper goes further and shows that the N-cornered hat method is potentially more powerful than multiple applications of the 3CH method to various combinations of N datasets. It allows the estimation of some of the error covariance terms, which can be chosen arbitrarily, but based on insights of which pairs of datasets are likely to be most independent (have smallest error covariances). The authors illustrate this with four different precipitation datasets.

It is important when describing the 3CH and 4CH to be very clear in describing the methods, notation, and equations, because the equations can quickly become confusing, especially to those unfamiliar with these relatively new methods. My major suggestion is to improve the description of the 3CH and 4CH method in Section 3.3. I found the discussion a bit unclear, in part due to the notation in Equations (1)-(3):

1.  There are currently two ways of writing the variance of the difference between two datasets. For example, in (1) var(A-B) is used, while in (2) and (3) $v_{A-B}$ is used. Please use one or the other for consistency.
2.  Also, $v_{A-t}$ is not defined, although one can assume it is the error variance of A. I suggest writing it as something like var($A_{err}$) or var($A\epsilon$), where $A_{err}$ or $\epsilon$ denotes random errors.
3.  More seriously, the covariance terms in (2) and (3) are not defined and could be misinterpreted. For example, $c_{AtBt}$ should be the covariance of the errors of A and B, but it could be misinterpreted as the covariance between the true values of A and B. Write something like cov($A\epsilon B\epsilon$) or cov($A_{err}B_{err}$).
4.  Finally, in Eqs. (2) and (3), each equation actually represents two equations. For example, the first equation represents the following two equations:
    $$V_{A-B} = V_{Aerr} + V_{Berr} - 2\text{cov}(A_{err}B_{err})$$
    $$V_{A-B} \cong V_{Aerr} + V_{Berr}$$

The second equation is only an approximation, and equality holds only when the error covariance term is zero.

In addition to the notation issues, I suggest a rewriting of lines 142-150 (I did not change the notation in this suggested revision):

*To that end, if we have a third independent observation system C, we can repeat the above steps with differences between A and C, and B and C to obtain a system of three equations and six unknowns (the error variances of the three datasets and three error covariance terms):*

$v_{A-B} = v_{A-t} + v_{B-t} - 2c_{AtBt}$
$v_{A-C} = v_{A-t} + v_{C-t} - 2c_{AtCt}$
$v_{B-C} = v_{B-t} + v_{C-t} - 2c_{BtCt}$        (2)

*If the three error covariance terms are small and can be neglected, the equations for the estimates of the true error variances can be solved from the remaining terms, which can be computed from the three collocated datasets.*

*In our case we have identified four observational systems, therefore we can constrain the computation a bit more. The four-cornered hat (4CH) method extends the system of equations with three more equations, (six in total, one for each pair of products), with 10 unknowns (the error variances of the four datasets and six error covariance terms (Eq. (3):*

$v_{A-B} = v_{A-t} + v_{B-t} - 2c_{AtBt}$
$v_{A-C} = v_{A-t} + v_{C-t} - 2c_{AtCt}$
$v_{A-D} = v_{A-t} + v_{D-t} - 2c_{AtDt}$
$v_{B-C} = v_{B-t} + v_{C-t} - 2c_{BtCt}$                         (3)
$v_{B-D} = v_{B-t} + v_{D-t} - 2c_{BtDt}$
$v_{C-D} = v_{C-t} + v_{D-t} - 2c_{CtDt}$

*If any four of the error covariance terms are assumed zero, we can solve for the error variance of the four datasets and the remaining two error covariance terms. Assessing which two covariance terms should be computed is a science-informed but subjective matter. The independence.....*

Minor comments:

1. The paper uses "data" as a singular noun. I realize many people use it this way, but technically "data" refers to more than one datum, and hence is plural. Consider changing all the "data is" to "data are".
2. Fig. 2—What is the circular feature over central Spain in three of the panels? And one panel in Fig. 4. A comment about this in the caption would be useful.
3. Line 175—It appears that the largest values of the differences between IMERG and ERA5 are along the coasts of western Europe and northern Italy (2-5 mm/day), not over the oceans where the largest magnitudes are 0.5-1.0 mm/day.
4. Both E-OBS and EOBS are used for the rain gauge dataset; please use one of these throughout. EOBS is used in many of the figures and in Line 180. E-OBS is used in many other places.
5. A. Line 183-It might be better to change the wording from "worst" and "best" to the less pejorative "highest" and "lowest" e.g. ranking the products from highest to lowest.
   B. Line 185-Same as A-highest instead of worst.

C. Line 191-Change "better" to "lower"

D. Line 245-Change "best to worst" to "lowest to highest."

6. Line 195—Sjoberg et al. (2021) discuss in some detail how negative error variances can be obtained and could be included here as a reference. Line 249 as well.
7. Are the units of Figs. 2, 4 and 5 $mm^2$ or $(mm/day)^2$ ?
8. Line 230-You could compute the error covariance of ERA5 and Opera in this case using the 4CH method, but it is not necessary.
9. Fig. A1 is outside Appendix A.
10. The error covariances shown in Fig. A2 are interesting, but it would also be interesting, and easier to interpret, if the corresponding error correlation map were shown.

---

## Referee Comment (RC2)

**General comment**

This study explores the estimation of random error variance in four gridded precipitation datasets sourced from various natural sources using the triangulation method called the Four Cornered Hat (4CH). The investigation focuses on Europe (over the intersection coverage of the precipitation products) and analyses the error variance patterns across different datasets. However, the overall outcome of the study is somewhat modest, resembling more of an experimental report rather than a comprehensive scientific paper. One notable concern is the ambiguity regarding whether the variant of the 3CH, the 4CH, represents an innovative aspect of this study. Furthermore, the introduction of the framework and methodology is brief and suffer from a lack of clarity, and the evaluation of the estimated variances is missing. In my assessment, the paper falls short of the standard expected by the Journal of HESS and would benefit significantly from a major revision. Below, I have listed my comments, which I believe will contribute to the preparation of an improved version of the manuscript.

**Major comments**

1) The model error hypothesis within the framework of the 3CH should be introduced more effectively. Specifically, the estimation of random error variance assumes additive noise $\varepsilon A$ (along with some unquantifiable bias bA) in the estimate A concerning the truth t:
A = t + bA + εA,

While this may seem self-evident, it's worth noting that this is not always the case, especially considering the further extension of triangulation techniques, such as Triple Collocation, where uncertainty estimation is developed for multiplicative error (Alemohammad et al., 2015). I believe this section would benefit from a clear introduction of the formalism utilized by the 3CH, including the notion and notation of the truth t, the estimates A, B, C, and D, as well as their respective errors, before presenting Equation 1.

*-Alemohammad, S. H., McColl, K. A., Konings, A. G., Entekhabi, D., and Stoffelen, A.: Characterization of precipitation product errors across the United States using multiplicative triple collocation, Hydrol. Earth Syst. Sci., 19, 3489–3503, https://doi.org/10.5194/hess-19-3489-2015, 2015.*

2) From my review, it remains unclear whether the variant 4CH introduced in the submitted manuscript represents a novelty.
a) If indeed it is a novel contribution, it is imperative to underscore this aspect within the manuscript. This could be achieved by delineating the limitations of the 3CH in the methods section, thereby emphasizing the innovation brought forth by the 4CH. Additionally, the title and abstract should be modified to highlight this advancement compared to existing state-of-the-art techniques. It's noteworthy to mention that extensions beyond three datasets have previously been explored within the framework of Triple Collocation (Pan et al., 2015; Gruber et al., 2016; Vogelzang, 2021). Therefore, the authors should reference these prior works while describing the 4CH approach.

b) If the variant 4CH has already been documented in the literature, this information must be explicitly stated, and the relevant references should be cited.

*-Gruber, A., Su, C.-H., Crow, W. T., Zwieback, S., Dorigo, W. A., & Wagner, W. (2016a). Estimating error cross-correlations in soil moisture data sets using extended collocation analysis. Journal of Geophysical Research: Atmospheres, 121, 1208–1219.* [https://doi.org/10.1002/2015JD024027](https://doi.org/10.1002/2015JD024027)
*-Pan, M., Fisher, C. K., Chaney, N. W., Zhan, W., Crow, W. T., Aires, F., et al. (2015). Triple collocation: Beyond three estimates and separation of structural/non-structural errors. Remote Sensing of Environment, 171, 299–310.* [https://doi.org/10.1016/j.rse.2015.10.028](https://doi.org/10.1016/j.rse.2015.10.028)
*-Vogelzang, J., & Stoffelen, A. (2021). Quadruple collocation analysis of in-situ, scatterometer, and NWP winds. Journal of Geophysical Research: Oceans, 126, e2021JC017189.* [https://doi.org/10.1029/2021JC017189](https://doi.org/10.1029/2021JC017189)

3)  The error covariance estimate (as illustrated in Figure A.2) should be incorporated into the main text of the article and deeper analyzed. For example, comparing it with the variance estimates (figure 2) could demonstrate to what extent neglecting these quantities in 3CH methods is accurate. Since these covariance estimates are among the primary outputs of the 4CH method, their inclusion and examination could significantly enhance the understanding and interpretation of the study's findings.

4) 3CH and 4CH method could be compared at least in the appendix or in the main text.

5) The study lacks in terms of evaluation. While I understand that the triangulation technique assumes the unavailability of ground truth, there are strategies that could be employed. For instance, using some gauge stations (not interpolated) from dense networks (over France and Germany) :
a) The author could analyze the impact of adding another dataset (e.g., EOBS) in the estimation of uncertainties (error variance), comparing results from Figure A3 and Figure 2. This comparative analysis would provide valuable insights into the effectiveness of incorporating additional datasets within the 4CH framework.
b) The author could evaluate the error variance pattern of EOBS with respect to the gauge density to investigate how its error variance is link to the interpolation itself.

Such an evaluation, albeit limited, would enhance the robustness of the proposed analysis.

6) In Figure 5, the quality issues observed for the OPERA dataset, as seen previously in Figure 2, almost disappeared in the winter/summer stratification. The author suggests that a plausible explanation for this phenomenon is the presence of outliers that are only present on certain days of April or October (therefore excluded in both composites), which exert a strong influence on the overall result. I recommend that the author thoroughly investigate this point and, if necessary, consider filtering out these outliers before estimating the error variance.

7) In Figures 4, 5, and A3, the gray area representing negative variance for one dataset should be masked for all the other three datasets in the figures. Furthermore, this discrepancy should be thoroughly discussed in the text as it underscores potential shortcomings in the 4CH approach.

**Other comments**

)Title: The title should introduce the notion of "variant 4CH" rather than the classical 3CH.

2)In the article structure, Section 3.3 Method could be separated from Section 3.1 Data to balance the length of each section.

3) Notation used in different equations should be harmonized.

4)Figures 2, 5, A2, and A3 show the boundaries of Europe and Africa outside the frame of the figures when the PDF is read locally.

[Figure]

**Figure 2.** Variance of the differences between pairs of products.

6) L112: "In particular, the OPERA radar data has been interpolated from a finer grid using a conservative interpolation". Please add the interpolation scheme that has been used here?

---

## Author Comment (AC1)

**Responses to reviewers**

**An intercomparison of four gridded precipitation products over Europe using an extension of the three-cornered-hat method**, by Llorenç Lledó, Thomas Haiden, and Matthieu Chevallier, submitted to HESS.

**Reviewer 1**

This is an excellent paper that applies the three-cornered hat (3CH) and four-cornered hat (4CH) methods to estimate the uncertainties (random error variances) of four precipitation datasets. It is acceptable for publication after the authors consider some relatively minor changes that would improve the clarity of the paper. My full review with suggested edits is included as a Supplement.

We want to thank the reviewer for the positive appreciation of this manuscript, and for taking the time to read it and suggest specific improvements.

This is an excellent paper that applies the three-cornered hat (3CH) and four-cornered hat (4CH) methods to estimate the uncertainties (random error variances) of four precipitation datasets. It is acceptable for publication after the authors consider some relatively minor changes that would improve the clarity of the paper.

A novel aspect of the paper is to use the 4CH method to compute two of the error covariance terms that must be neglected in 3CH error variance estimates. Sjoberg et al. (2021) show that N datasets lead to (N - 1)(N - 2)/2 unique error variance estimates for each dataset when using the 3CH method. Furthermore, under the assumption of zero error covariances among all the datasets, a single error estimate for each dataset can be computed using all N datasets simultaneously (N-cornered hat method, a generalization of the 3CH method), which is identical to the mean of all the individual estimates. This paper goes further and shows that the N-cornered hat method is potentially more powerful than multiple applications of the 3CH method to various combinations of N datasets. It allows the estimation of some of the error covariance terms, which can be chosen arbitrarily, but based on insights of which pairs of datasets are likely to be most independent (have smallest error covariances). The authors illustrate this with four different precipitation datasets.

Thanks for providing this explanation of how N datasets were employed in Sjoeberg et al 2021. We have included this information in a new paragraph at the end of section 3.4, where we describe the novelty of our extension of the 3CH.

It is important when describing the 3CH and 4CH to be very clear in describing the methods, notation, and equations, because the equations can quickly become confusing, especially to those unfamiliar with these relatively new methods. My major suggestion is to improve the description of the 3CH and 4CH method in Section 3.3. I found the discussion a bit unclear, in part due to the notation in Equations (1)-(3):

1. There are currently two ways of writing the variance of the difference between two datasets. For example, in (1) var(A-B) is used, while in (2) and (3) vA-B is used. Please use one or the other for consistency.
2. Also, vA-t is not defined, although one can assume it is the error variance of A. I suggest writing it as something like var(Aerr) or var(Aϵ), where Aerr or ϵ denotes random errors.
3. More seriously, the covariance terms in (2) and (3) are not defined and could be misinterpreted. For example, cAtBt should be the covariance of the errors of A and B, but it could be misinterpreted as the covariance between the true values of A and B. Write something like cov(AϵBϵ) or cov(AerrBerr).
4. Finally, in Eqs. (2) and (3), each equation actually represents two equations. For example, the first equation represents the following two equations:
   VA-B = VAerr + VBerr – 2cov(AerrBerr)
   VA-B @ VAerr + VBerr
   The second equation is only an approximation, and equality holds only when the error covariance term is zero.

We agree that the notation was not carefully defined in the first manuscript. We have now defined $E\_A=A-t$ as the additive error of product A, and explicitly written all variance and covariance as Var(x) or Cov(x,y). We have also stated equation (1) as two separate equations and stated the system of equations (2) as two separate systems of equations, adding more clarity to the discussion. Equation (3) has been moved to a new subsection to present the four-cornered-hat methodology extension we employ.

In addition to the notation issues, I suggest a rewriting of lines 142-150 (I did not change the notation in this suggested revision):

*To that end, if we have a third independent observation system C, we can repeat the above steps with differences between A and C, and B and C to obtain a system of three equations and six unknowns (the error variances of the three datasets and three error covariance terms):*

*vA−B = vA−t +vB−t −2cAtBt*
*vA−C = vA−t +vC−t −2cAtCt*
*vB−C = vB−t +vC−t −2cBtCt*

*If the three error covariance terms are small and can be neglected, the equations for the estimates of the true error variances can be solved from the remaining terms, which can be computed from the three collocated datasets.*

*In our case we have identified four observational systems, therefore we can constrain the computation a bit more. The four-cornered hat (4CH) method extends the system of equations with three more equations, (six in total, one for each pair of products), with 10 unknowns (the error variances of the four datasets and six error covariance terms (Eq. (3):*

*vA−B = vA−t +vB−t −2cAtBt*

$$vA{-}C = vA{-}t + vC{-}t - 2cAtCt$$
$$vA{-}D = vA{-}t + vD{-}t - 2cAtDt$$
$$vB{-}C = vB{-}t + vC{-}t - 2cBtCt$$
$$vB{-}D = vB{-}t + vD{-}t - 2cBtDt$$
$$vC{-}D = vC{-}t + vD{-}t - 2cCtDt$$

*If any four of the error covariance terms are assumed zero, we can solve for the error variance of the four datasets and the remaining two error covariance terms. Assessing which two covariance terms should be computed is a science-informed but subjective matter. The independence.....*

We have followed this rationale in the reorganization of section 3.3. In response to reviewer 2 we have divided the section in two, with the 4CH method being presented in new section 3.4. Thanks for taking the time to suggest a clearer way to present the equations.

**Minor comments:**

1. The paper uses "data" as a singular noun. I realize many people use it this way, but technically "data" refers to more than one datum, and hence is plural. Consider changing all the "data is" to "data are".
   Unless the editor has a strong opinion about this, we would prefer to use this generally accepted singular usage.

2. Fig. 2—What is the circular feature over central Spain in three of the panels? And one panel in Fig. 4. A comment about this in the caption would be useful.
   This pattern is related to quality issues in the OPERA dataset, specifically the weather radar in Madrid has some strong outliers in specific dates. Those are discussed in line 215 and also later in 236 of the annotated manuscript. We believe it is a great illustration of the power of the 3CH method, and hence decided to leave this visible. It is not the intent of this work to curate specific datasets (i.e. apply additional quality controls, post-process, calibrate) but rather estimate its usefulness for forecast verification. We have added a note on the figure caption, as suggested.

3. Line 175—It appears that the largest values of the differences between IMERG and ERA5 are along the coasts of western Europe and northern Italy (2-5 mm/day), not over the oceans where the largest magnitudes are 0.5-1.0 mm/day.
   We realize that this sentence was difficult to follow. We wanted to describe the behavior of the products over the ocean. We have changed the sentence to: "The largest differences can be seen over the ocean, where OPERA is drier than IMERG and ERA5 by 2 to 5 mm, and IMERG is wetter than ERA5 by 1 to 2 mm".

4. Both E-OBS and EOBS are used for the rain gauge dataset; please use one of these throughout. EOBS is used in many of the figures and in Line 180. E-OBS is used in many other places.
   Thanks for pointing this out. We have corrected EOBS to E-OBS everywhere, except in the figures that show differences of products, where we wrote E_OBS to differentiate clearly the two products being subtracted.

5.
   A. Line 183-It might be better to change the wording from "worst" and "best" to the less pejorative "highest" and "lowest" e.g. ranking the products from highest to lowest.
   B. Line 185-Same as A-highest instead of worst.
   C. Line 191-Change "better" to "lower"
   D. Line 245-Change "best to worst" to "lowest to highest."
   Agreed and changed. We also changed the legend of fig. A1 accordingly.
6. Line 195—Sjoberg et al. (2021) discuss in some detail how negative error variances can be obtained and could be included here as a reference. Line 249 as well.
   We have included a reference to section 4e of Sjoberg et al. (2021), and also a reference to section 2.5 of Pan et al. (2015) which also contains a good theoretical basis for understanding when negative error variances happen.
7. Are the units of Figs. 2, 4 and 5 $mm^2$ or $(mm/day)^2$ ?
   Total precipitation is measured in mm, and its variance in mm^2. Figure 2 showed mm/day which was erroneous and has been corrected. We also corrected the sentences of section 4 where this figure is discussed.
8. Line 230-You could compute the error covariance of ERA5 and Opera in this case using the 4CH method, but it is not necessary.
   We agree that it is not impossible to find plausible assumptions for running the 4CH method over the US, but we haven't explored that avenue in depth.
9. Fig. A1 is outside Appendix A.
   This was handled by the journal latex template. I would expect the copy-edition to fix that. Anyways, after revising the manuscript the figure is placed in the right place.
10. The error covariances shown in Fig. A2 are interesting, but it would also be interesting, and easier to interpret, if the corresponding error correlation map were shown
    We have produced error correlation maps, and we present them in the appendix. The covariances have been moved to the main text as a response to a suggestion by reviewer 2. One particular problem with the error correlations, though, is that computing them from estimated covariance and variances does not guarantee that the result is in the [-1,1] range, and the occurrence of spurious zero or negative variances leads to undefined values. This has been mentioned in the Appendix.

---

## Author Comment (AC2)

**Responses to reviewers**

**An intercomparison of four gridded precipitation products over Europe using an extension of the three-cornered-hat method**, by Llorenç Lledó, Thomas Haiden, and Matthieu Chevallier, submitted to HESS.

**Reviewer 2**

This study explores the estimation of random error variance in four gridded precipitation datasets sourced from various natural sources using the triangulation method called the Four Cornered Hat (4CH). In my assessment, the paper falls short of the standard expected by the Journal of HESS and would benefit significantly from a major revision. My main concern is the ambiguity regarding whether the variant of the 3CH, the 4CH, is an innovative aspect of this study. My full review is included as a Supplement.

We want to thank the reviewer for taking the time to read the manuscript and suggest aspects that require improvement. In the revised manuscript we have clarified the novelty of our method by putting it in context with existing multiple-collocation techniques.

**General comment**

This study explores the estimation of random error variance in four gridded precipitation datasets sourced from various natural sources using the triangulation method called the Four Cornered Hat (4CH). The investigation focuses on Europe (over the intersection coverage of the precipitation products) and analyses the error variance patterns across different datasets.

However, the overall outcome of the study is somewhat modest, resembling more of an experimental report rather than a comprehensive scientific paper. One notable concern is the ambiguity regarding whether the variant of the 3CH, the 4CH, represents an innovative aspect of this study. Furthermore, the introduction of the framework and methodology is brief and suffer from a lack of clarity, and the evaluation of the estimated variances is missing. In my assessment, the paper falls short of the standard expected by the Journal of HESS and would benefit significantly from a major revision. Below, I have listed my comments, which I believe will contribute to the preparation of an improved version of the manuscript.

We agree that the methods section of the original manuscript lacked clarity and context and we have worked to improve this aspect in the revised manuscript. Regarding the modesty of the results, not for being concise a study should be dismissed of its value. In this case, the results we present are very relevant for guidance on forecast verification of precipitation. The fact that conclusions can be extracted from a limited number of backed assumptions is a strength rather than a weakness. We have been very careful not to run too many analyses based on different assumptions and then try to reconcile the results. Instead, we believe the best we can do is select a careful set of assumptions and discuss the results as the best estimates we can obtain from the data at hand.

**Major comments**

1) The model error hypothesis within the framework of the 3CH should be introduced more effectively. Specifically, the estimation of random error variance assumes additive noise εA (along with some unquantifiable bias bA) in the estimate A concerning the truth t:

A = t + bA + εA,

While this may seem self-evident, it's worth noting that this is not always the case, especially considering the further extension of triangulation techniques, such as Triple Collocation, where uncertainty estimation is developed for multiplicative error (Alemohammad et al., 2015). I believe this section would benefit from a clear introduction of the formalism utilized by the 3CH, including the notion and notation of the truth t, the estimates A, B, C, and D, as well as their respective errors, before presenting Equation 1.

- Alemohammad, S. H., McColl, K. A., Konings, A. G., Entekhabi, D., and Stoffelen, A.: Characterization of precipitation product errors across the United States using multiplicative triple collocation, Hydrol. Earth Syst. Sci., 19, 3489–3503, https://doi.org/10.5194/hess-19-3489-2015, 2015.

We agree that nomenclature was not clear enough in the first manuscript, and the methodology lacked clarity. The only error hypothesis we use is that errors are defined additively, by setting E_A=A-t as the error of product A. This is the usual hypothesis in forecast verification. We have included this information in the revised manuscript, and we have enhanced the notation of sections 3.3 and 3.4 and explained in more detail the equations. The relationship of the 3CH method with the triple collocation method is fully described by Sjoberg et al. 2021 and there is no need to repeat those considerations here. The 3CH methodology is well aligned with our hypothesis and hence suitable here.

2) From my review, it remains unclear whether the variant 4CH introduced in the submitted manuscript represents a novelty.

The main point of the manuscript is to compare the uncertainties of four different precipitation datasets, and this is done by using an extension of the 3CH method, the 4CH method. Sjoberg et al. (2021) already introduced the notion of the N-CH analysis, but in that case, they use the additional datasets to compute multiple 3CH estimates from triplets. As far as the authors know, using the additional dataset to explicitly compute some of the 3CH covariance terms is novel, but it is not the main point of the paper. We do not believe this is a substantial modification of the original method that deserves special attention. We simply adapted the existing tools to the specific characteristics of the available datasets.

a) If indeed it is a novel contribution, it is imperative to underscore this aspect within the manuscript. This could be achieved by delineating the limitations of the 3CH in the methods section, thereby emphasizing the innovation brought forth by the 4CH. Additionally, the title and abstract should be modified to highlight this advancement compared to existing state-of-the-art techniques. It's noteworthy to mention that extensions beyond three datasets have previously been explored within the framework of Triple Collocation (Pan et al., 2015; Gruber et al., 2016; Vogelzang, 2021). Therefore, the authors should reference these prior works while describing the 4CH approach.

We agree with these suggestions. We have divided section 3.3 into one part for the 3CH and one part presenting the 4CH extension in the revised manuscript. We have also modified the title, and included a paragraph where we discuss the novelty of our method in relation to the existing literature. We have been cautious of not mixing 3CH and TC in the methodology section, which have different assumptions, but we have otherwise cited these studies because the rationale is similar.

b) If the variant 4CH has already been documented in the literature, this information must be explicitly stated, and the relevant references should be cited.

- Gruber, A., Su, C.-H., Crow, W. T., Zwieback, S., Dorigo, W. A., & Wagner, W. (2016a). Estimating error cross-correlations in soil moisture data sets using extended collocation analysis. Journal of Geophysical Research: Atmospheres, 121, 1208–1219. https://doi.org/10.1002/2015JD024027
- Pan, M., Fisher, C. K., Chaney, N. W., Zhan, W., Crow, W. T., Aires, F., et al. (2015). Triple collocation: Beyond three estimates and separation of structural/non-structural errors. Remote Sensing of Environment,171, 299–310. https://doi.org/10.1016/j.rse.2015.10.028
- Vogelzang, J., & Stoffelen, A. (2021). Quadruple collocation analysis of in-situ, scatterometer, and NWP winds. Journal of Geophysical Research: Oceans, 126, e2021JC017189. https://doi.org/10.1029/2021JC017189

Thanks for providing these interesting references. They all refer to TC and not to 3CH (albeit Pan et al, indeed present the 3CH equations under the TC name). Indeed Volgelzang et al. use a very similar concept to what we present, but with TC. As far as the authors know, this is the first time the idea is used for the 3CH. We have included these references and also described the relationship with our extension at the end of the new 4CH sub-section.

3) The error covariance estimate (as illustrated in Figure A.2) should be incorporated into the main text of the article and deeper analyzed. For example, comparing it with the variance estimates (figure 2) could demonstrate to what extent neglecting these quantities in 3CH methods is accurate. Since these covariance estimates are among the primary outputs of the 4CH method, their inclusion and examination could significantly enhance the understanding and interpretation of the study's findings.

We have moved this figure to the main text and added a description of it. Also, in reaction to a suggestion from reviewer 1, we have included error correlations in the appendix, which might be easier to interpret in some cases. Indeed, the areas with negative variances are collocated with large error covariances, and we discuss this in the text.

4) 3CH and 4CH method could be compared at least in the appendix or in the main text.

We have included a brief discussion of the differences between the 3CH and 4CH results in the appendix. We do not see large differences, except a slightly lower error variance for IMERG with the assumption of uncorrelated errors. We believe the assumptions for the 4CH analysis are better than those for the 3CH and therefore those results should be superior.

5) The study lacks in terms of evaluation. While I understand that the triangulation technique assumes the unavailability of ground truth, there are strategies that could be employed. For instance, using some gauge stations (not interpolated) from dense networks (over France and Germany) :
E-OBS is very efficient in making use of as many stations as possible from European high-density networks, and there is no such independent set available for verification, as far as the authors know. KNMI states in the E-OBS product documentation that: "For a considerable number of countries the number of stations used is the complete national network and therefore much more dense than the station network that is routinely shared among NMHSs (which is the basis of other gridded datasets). The density of stations gradually increases through collaborations with NMHSs within European research contracts."
a) The author could analyze the impact of adding another dataset (e.g., EOBS) in the estimation of uncertainties (error variance), comparing results from Figure A3 and Figure 2. This comparative analysis would provide valuable insights into the effectiveness of incorporating additional datasets within the 4CH framework.
Please see our answer to point 4) above. The increase in the variance estimates not only comes from having more datasets but also from having better assumptions on the error correlations.
b) The author could evaluate the error variance pattern of EOBS with respect to the gauge density to investigate how its error variance is link to the interpolation itself. Such an evaluation, albeit limited, would enhance the robustness of the proposed analysis.
We do not have access to the underlying station density of E-OBS, unfortunately. The product documentation (https://confluence.ecmwf.int/display/CKB/E-OBS+daily+gridded+observations+for+Europe+from+1950+to+present%3A+Product+user+guide#EOBSdailygriddedobservationsforEuropefrom1950topresent:Productuserguide-Theunderlyingstationdataset) shows this plot for v20 of the product, but we use v26 so probably the situation changed in some countries.

[Figure]

6) In Figure 5, the quality issues observed for the OPERA dataset, as seen previously in Figure 2, almost disappeared in the winter/summer stratification. The author suggests that a plausible explanation for this phenomenon is the presence of outliers that are only present on certain days of April or October (therefore excluded in both composites), which exert a strong influence on the overall result. I recommend that the author thoroughly investigate this point and, if necessary, consider filtering out these outliers before estimating the error variance.

We believe it is a great illustration of the power of the 3CH method, and hence decided to leave this visible. It is not the intent of this work to curate specific datasets (i.e. apply additional quality controls, post-process, calibrate) but rather estimate their usefulness for forecast verification.

7) In Figures 4, 5, and A3, the gray area representing negative variance for one dataset should be masked for all the other three datasets in the figures. Furthermore, this discrepancy should be thoroughly discussed in the text as it underscores potential shortcomings in the 4CH approach.

Masking the grid-points in which one estimate is negative would hide which of the datasets received the negative value. The numbers, albeit not exact, are still indicative of the product quality. These negative values were already discussed in section 4 of the original manuscript. The negative variances can only be seen over a small portion of Europe and are due to the fact the error correlation assumptions are not correct in those grid points. In the revised manuscript we have backed this sentence by the inclusion of references discussing when those negative values occur, (in particular, those are described in section 4e of Sjoberg et al (2021) and section 2.5 of Pan et al. (2015)).

**Other comments**
1)Title: The title should introduce the notion of "variant 4CH" rather than the classical 3CH.

We have modified the title to highlight that we use an extension of the 3CH method.

2)In the article structure, Section 3.3 Method could be separated from Section 3.1 Data to balance the length of each section.
We have separated the 3CH and 4CH description into two subsections, resulting in better balanced sections.

3) Notation used in different equations should be harmonized.
Agreed and changed. This is in line with comments from reviewer 1 also.

4)Figures 2, 5, A2, and A3 show the boundaries of Europe and Africa outside the frame of the figures when the PDF is read locally.
We have updated the software stack and now used a more modern version of python, xarray and matplotlib to recreate the figures. However, we could not check this particular issue because the figures look good in Adobe Acrobat, and we could not reproduce the problem.

6) L112: "In particular, the OPERA radar data has been interpolated from a finer grid using a conservative interpolation". Please add the interpolation scheme that has been used here?
The OPERA data has been upscaled with the gdalwarp utility, and in particular the "average" method, which according to the documentation (https://gdal.org/programs/gdalwarp.html) "computes the weighted average of all non-NODATA contributing pixels". We have included this information in the revised manuscript in the Data collocation section.

---

## Referee Report (RR1)

22 July 2024

Review of revised version of *An intercomparison of four gridded precipitation products over Europe using the three-cornered-hat method* by L. Lledó, T. Haiden, and M. Chevallier for EGUsphere.

Richard Anthes
anthes@ucar.edu

The revised paper is an improvement over an already excellent paper and is acceptable for publication. The authors have responded adequately to my first review and the paper is clearer as a result. I have only a few very minor suggestions for the final version. I do not need to see it again unless there are specific questions or concerns.

Minor comments:

1. The notation $\binom{N}{2}$ was unfamiliar to me. The factorial form would be more understandable to all readers.
2. I am not sure Eq. 4 is necessary. Also, the { in Eqs. (3)-(5) are not necessary.
3. In line 178, the two datasets for which the error covariances are not assumed zero and instead calculated (IMERG and Opera) should be define explicitly. You could write: Therefore, we have decided to explicitly compute the error covariance between IMERG and Opera.
4. Shouldn't the *Author Contributions, Competing interests*, and *Acknowledgement*s come before the Appendix, right after *Code and data availability*?

---

## Referee Report (RR2)

**General comment**

This study examines the estimation of random error variance in four gridded precipitation datasets derived from various sources using a triangulation method known as the Four Cornered Hat (4CH). The research focuses on Europe (within the overlapping coverage area of the precipitation products) and analyzes the error variance patterns across the different datasets.

The authors have improved the logic and structure of their article while positively addressing my main concern regarding the innovation of the study: their extension of the 3CH method to the 4CH method is built on the work of Sjoberg et al. (2021). But they used additional datasets to compute two of the error covariance terms that must have been neglected in 3CH framework, which is the novel aspect of their work. However, I do not understand why the authors chose to highlight discussion on number that might be relevant for guidance on precipitation forecast verification but are not evaluated or validated, instead of better emphasizing the novelty of their methodological approach. That said, the paper, in its new structure, is acceptable for publication.

Nevertheless, I offer here some responses and clarifications on the authors' answers that might be considered to further improve the paper.

**Comment**

1) Regarding the evaluation, I acknowledge that the authors present valuable points showing that neither a) in situ network data nor b) E-OBS gauge density information is available for validating their error variance estimates. The authors must be aware that these two kinds of independent validation datasets were given as examples. It is the authors' responsibility to find a way to validate their analysis results to some extent. If this is not possible, I suggest explicitly discussing in the conclusion that validating such numbers is challenging given the current state of the art.

2) Regarding my comment suggesting masking (in grey) pixels where one of the datasets has negative variance in the 4CH method, the authors state in their response: "The numbers, albeit not exact, are still indicative of the product quality." From my point of view, deriving negative variance for at least one dataset indicates that the 4CH hypotheses do not hold at these particular locations. Since the variance (and covariance) are co-estimated dependently, I would not trust any of these estimates where one of them is incorrect. How can negative variance be linked to product quality? I suggest adding to the manuscript a short message noting that if one of the error estimates is non-physical, the other five estimates at the same location must be treated with caution.

3) Regarding my comment on the disappearance of quality issues observed for the OPERA dataset in the winter/summer stratification, the authors claim that this is a great illustration of the power of the 3CH. I respectfully disagree with this point because, in the conclusion, the authors' final guidance for choosing the verification dataset relies on the non-stratified variance estimates. In the authors' logic OPERA is less useful all year long due to only "a few outliers only present on some days in April or October", which does not seem accurate to me. The fact that a few outliers strongly influence the overall variance estimate is one of the limit of the current 4CH approach. On the contrary, I suggest the authors emphasize the power of their stratification to move from "static" error variance estimates toward state-dependent error variance estimates. Note that such an approach has been recently introduced in the framework of the Triple Collocation (Pellet et al. 2022).

References :
- Sjoberg, J. P., Anthes, R. A., and Rieckh, T.: The Three-Cornered Hat Method for Estimating Error Variances of Three or More Atmospheric Datasets. Part I: Overview and Evaluation, Journal of Atmospheric and Oceanic Technology, 38, 555–572, https://doi.org/10.1175/JTECH-380D-19-0217.1, 2021.
- V. Pellet, "A Binned Triple Collocation for Estimating Regime-Dependent Uncertainties of Precipitation," in *IEEE Geoscience and Remote Sensing Letters*, vol. 19, pp. 1-5, 2022, Art no. 1005405, doi: 10.1109/LGRS.2022.3162703

---

## Author Response (AR2)

**Responses to reviewers**

**An intercomparison of four gridded precipitation products over Europe using an extension of the three-cornered-hat method**, by Llorenç Lledó, Thomas Haiden, and Matthieu Chevallier, submitted to HESS.

**Reviewer 1**

The revised paper is an improvement over an already excellent paper and is acceptable for publication. The authors have responded adequately to my first review and the paper is clearer as a result. I have only a few very minor suggestions for the final version. I do not need to see it again unless there are specific questions or concerns.

Minor comments:

1. The notation $\binom{N}{2}$ was unfamiliar to me. The factorial form would be more understandable to all readers.

The notation $\binom{N}{2}$ denotes the binomial coefficient N over 2 or N choose 2 and, as far as the authors know, is a standard mathematical notation in the field of combinatorics, and more compact than the expanded form with three factorial terms. Unless the journal editor has a clear requirement, authors would prefer to keep this compact form.

2. I am not sure Eq. 4 is necessary. Also, the { in Eqs. (3)-(5) are not necessary.

We have included those brackets to denote a system of equations. This is also standard mathematical notation as far as authors know. Unless the journal editor has a clear requirement, authors would prefer to keep these brackets.

3. In line 178, the two datasets for which the error covariances are not assumed zero and instead calculated (IMERG and Opera) should be define explicitly. You could write:

Therefore, we have decided to explicitly compute the error covariance between IMERG and Opera.

Thanks for this suggestion. We have added an additional sentence for each bullet point where we explicitly write the corresponding error covariance term, and whether we set it to zero or we have computed it.

4. Shouldn't the Author Contributions, Competing interests, and Acknowledgements come before the Appendix, right after Code and data availability?

We have employed the Latex journal template provided by the journal, and this is the order that appears on compilation.  If this is an issue, I would expect the copyediting to fix it.

**An intercomparison of four gridded precipitation products over Europe using an extension of the three-cornered-hat method**, by Llorenç Lledó, Thomas Haiden, and Matthieu Chevallier, submitted to HESS.

**Reviewer 2**

**General comment**

This study examines the estimation of random error variance in four gridded precipitation datasets derived from various sources using a triangulation method known as the Four Cornered Hat (4CH). The research focuses on Europe (within the overlapping coverage area of the precipitation products) and analyzes the error variance patterns across the different datasets.

The authors have improved the logic and structure of their article while positively addressing my main concern regarding the innovation of the study: their extension of the 3CH method to the 4CH method is built on the work of Sjoberg et al. (2021). But they used additional datasets to compute two of the error covariance terms that must have been neglected in 3CH framework, which is the novel aspect of their work. However, I do not understand why the authors chose to highlight discussion on number that might be relevant for guidance on precipitation forecast verification but are not evaluated or validated, instead of better emphasizing the novelty of their methodological approach. That said, the paper, in its new structure, is acceptable for publication.

Nevertheless, I offer here some responses and clarifications on the authors' answers that might be considered to further improve the paper.

**Comment**

1) Regarding the evaluation, I acknowledge that the authors present valuable points showing that neither a) in situ network data nor b) E-OBS gauge density information is available for validating their error variance estimates. The authors must be aware that these two kinds of independent validation datasets were given as examples. It is the authors' responsibility to find a way to validate their analysis results to some extent. If this is not possible, I suggest explicitly discussing in the conclusion that validating such numbers is challenging given the current state of the art.

Authors do not want to set aside an independent dataset for verification (which would already imply a judgment of the verification data quality) and prefer to treat all data available with the 4CH method directly. With this, we have been able to show that radar estimates are sometimes (e.g. in Germany) of better quality than station-derived estimates. We have added a sentence in the conclusions stating that the observational error variances obtained are only accurate if the assumptions of the 4CH analysis are fulfilled:

*"All the conclusions above are based on estimates of error variance obtained with an extension of the three-cornered-hat method, which in turn relies on a set of*

*error covariance assumptions. Those assumptions were carefully selected based on physical considerations of the different observational systems but have not been independently validated. The error variance estimates can differ from the true error variances whenever the error orthogonality assumptions are not fulfilled."*

2) Regarding my comment suggesting masking (in grey) pixels where one of the datasets has negative variance in the 4CH method, the authors state in their response: "The numbers, albeit not exact, are still indicative of the product quality." From my point of view, deriving negative variance for at least one dataset indicates that the 4CH hypotheses do not hold at these particular locations. Since the variance (and covariance) are co-estimated dependently, I would not trust any of these estimates where one of them is incorrect. How can negative variance be linked to product quality? I suggest adding to the manuscript a short message noting that if one of the error estimates is non-physical, the other five estimates at the same location must be treated with caution.

We have added a sentence in the conclusions stating that the observational error variances obtained are only accurate if the assumptions of the 4CH analysis are fulfilled (see above). Authors believe those assumptions are strong and valid for most of the grid points.

3) Regarding my comment on the disappearance of quality issues observed for the OPERA dataset in the winter/summer stratification, the authors claim that this is a great illustration of the power of the 3CH. I respectfully disagree with this point because, in the conclusion, the authors' final guidance for choosing the verification dataset relies on the non-stratified variance estimates. In the authors' logic OPERA is less useful all year long due to only "a few outliers only present on some days in April or October", which does not seem accurate to me. The fact that a few outliers strongly influence the overall variance estimate is one of the limit of the current 4CH approach. On the contrary, I suggest the authors emphasize the power of their stratification to move from "static" error variance estimates toward state-dependent error variance estimates. Note that such an approach has been recently introduced in the framework of the Triple Collocation (Pellet et al. 2022).

Authors do not think that OPERA is less useful all year long. Indeed, the authors final guidance in the conclusions (line 240) already reads: *"The OPERA data is a higher resolution product, and although it might be biased with respect to in-situ measurements, with a bit of additional quality control it might be the best way to go for verifying high-resolution simulations over Europe".*

References :
- Sjoberg, J. P., Anthes, R. A., and Rieckh, T.: The Three-Cornered Hat Method for Estimating Error Variances of Three or More Atmospheric Datasets. Part I: Overview and Evaluation, Journal of Atmospheric and Oceanic Technology, 38, 555–572, https://doi.org/10.1175/JTECH-380D-19-0217.1, 2021.
- V. Pellet, "A Binned Triple Collocation for Estimating Regime-Dependent Uncertainties of Precipitation," in IEEE Geoscience and Remote Sensing Letters, vol. 19, pp. 1-5, 2022, Art no. 1005405, doi: 10.1109/LGRS.2022.3162703